# Epigenetic Mechanisms in Developmental Alcohol-Induced Neurobehavioral Deficits

**DOI:** 10.3390/brainsci6020012

**Published:** 2016-04-08

**Authors:** Balapal S. Basavarajappa, Shivakumar Subbanna

**Affiliations:** 1Division of Analytical Psychopharmacology, Nathan Kline Institute for Psychiatric Research, 140 Old Orangeburg Road, Orangeburg, NY 10962, USA; SShivakumar@nki.rfmh.org; 2New York State Psychiatric Institute, New York, NY 10032, USA; 3Department of Psychiatry, College of Physicians & Surgeons, Columbia University, New York, NY 10032, USA

**Keywords:** FASD, Synaptic plasticity, Learning and memory, FAS, DNA and histone modification

## Abstract

Alcohol consumption during pregnancy and its damaging consequences on the developing infant brain are significant public health, social, and economic issues. The major distinctive features of prenatal alcohol exposure in humans are cognitive and behavioral dysfunction due to damage to the central nervous system (CNS), which results in a continuum of disarray that is collectively called fetal alcohol spectrum disorder (FASD). Many rodent models have been developed to understand the mechanisms of and to reproduce the human FASD phenotypes. These animal FASD studies have provided several molecular pathways that are likely responsible for the neurobehavioral abnormalities that are associated with prenatal alcohol exposure of the developing CNS. Recently, many laboratories have identified several immediate, as well as long-lasting, epigenetic modifications of DNA methylation, DNA-associated histone proteins and microRNA (miRNA) biogenesis by using a variety of epigenetic approaches in rodent FASD models. Because DNA methylation patterns, DNA-associated histone protein modifications and miRNA-regulated gene expression are crucial for synaptic plasticity and learning and memory, they can therefore offer an answer to many of the neurobehavioral abnormalities that are found in FASD. In this review, we briefly discuss the current literature of DNA methylation, DNA-associated histone proteins modification and miRNA and review recent developments concerning epigenetic changes in FASD.

## 1. Introduction

A change of gene expression evoked by mechanisms other than alterations in DNA sequence contributes to the advancement of cell and tissue development and significantly aids in the maintenance of a regular cell-specific function. Thus, dynamically regulated, epigenetic mechanisms differing throughout life and between various cell populations have been shown to contribute to the development of neuronal circuits and brain function. The major epigenetic alterations that are known to regulate gene expression in the brain are as follows: (1) DNA methylation of cytosine residues in promoter-rich cytosine phosphate guanine (CpG) islands; (2) acetylation, methylation and other covalent posttranslational modification (PMT) of DNA-associated histone proteins; (3) chromatin remodeling proteins that indirectly affect gene transcription; (4) pre-mRNA editing and splicing by non-coding, small nucleolar RNA (snoRNAs); (5) mRNA processing, translation, and stability of binding proteins, long non-coding RNAs (lncRNAs) and microRNAs (miRNAs); and (6) mRNA translation that is controlled by cellular signaling molecules. These epigenetic mechanisms can selectively respond to environmental factors and alter transcriptional machinery leading to altered physiological and pathological processes in the brain. An increasing number of environmental factors, such as maternal care, stress, chemical agents, drugs of abuse, alcohol and nutritional disparity [1,2,3,4,5,6,7,8], are undoubtedly shown to remodel epigenetics. However, the mechanism by which epigenetic remodeling events are linked to environmentally induced diseases is poorly understood. The review will provide a current understanding of epigenetic mechanisms and how alcohol exposure during development, as a type of environmental teratogen, can influence epigenetic programs to alter gene transcription and induce persistent early alcohol-associated abnormalities.

## 2. Epigenetic Mechanisms

### 2.1. DNA Methylation

Our modern understanding of epigenetic marks on DNA has provided insight into an additional regulatory system in the cell that operates to regulate when and where a particular gene should be expressed during development and normal cellular homeostasis. Historically, the presence of methylated DNA was identified in mammals as soon as DNA was recognized as the primary genetic material [9,10]. It was in 1948 when the presence of modified cytosine in the calf thymus and the existence of this modified cytosine naturally in DNA was established [11]. Later, studies have confirmed that methylation of cytosine (DNA methylation) regulate gene activity during development [12,13]. Currently, DNA methylation along with other remodeling factors is considered to be the major epigenetic remodeling event known to control gene activities (for references see [14,15]).

The methylation of cytosine (mC) is a covalent modification of DNA (Figure 1). A family of DNA methyltransferases (DNMTs) transfer a methyl group from S-adenyl methionine (SAM) to the cytosine residue to form 5-mC [16,17]. DNMT3a and -3b are characterized as *de novo* DNMTs partly because they can establish a new methylation pattern for unmodified DNA. Conversely, DNMT1 copies the DNA methylation pattern from the parental DNA strand onto the newly produced daughter DNA strand during DNA replication [18]. These unique functions of DNMTs supposedly ensure that the DNA methylation pattern is maintained and preserved in a tissue-specific manner across different individuals [19,20]. All three DNMTs are expressed extensively during embryo development, and a significant level is found in postmitotic neurons in the mature mammalian brain [21,22,23,24,25]. These observations suggest that DNMTs and DNA methylation patterns have an important novel role not only in early development but also in mature brain function [26,27]. Additionally, several recent studies have described at least two mechanisms that actively remove mC. One mechanism is through the activity of deaminases that catalyze the conversion of mC to thymidine [28]. A second mechanism is through the action of ten-eleven translocation *(*TET*)* proteins **(**α-ketoglutarate-dependent dioxygenases). TET proteins oxidize 5-mC to 5-hydroxymethylcytosine (5-hmC) using an oxygen- and α-ketoglutarate-dependent mechanism. This is then oxidized to 5-formylcytosine and finally to 5-carboxylcytosine (5-caC), which is then followed by the removal of the modified base through base excision repair and glycosylase activity [15,28]. DNA demethylation processes via 5-hmC were shown to operate in both developmental, as well as in an age-dependent manner in the mammalian brain [29], thereby providing the basis for an important epigenetic regulator of gene expression [30]. These discoveries suggest that external factors, such as environmental exposure/experiences, have the ability to modify the DNA methylation pattern. Indeed, research over the last decade suggests that DNA methylation responds to environmental exposure/experience, thereby resulting in stable phenotypes [31,32,33]. It has been suggested that these additional changes in DNA methylation that are due to environmental influences may then be inherited in a transgenerational manner [34,35]. The vast majority of recent data suggest that rapid and dynamic methylation and demethylation of specific genes in the brain have a direct role in synaptic plasticity, learning and memory formation [14,15,26,36,37,38,39,40].

Another group of proteins that work closely with methylated DNA to control gene transcription in the brain is the family of methyl-binding proteins. Methyl-binding proteins are persistently expressed in the adult brain and often act as gene repressors through binding to methylated cytosines [47,48]. The methyl-binding protein 2 (MeCP2) is expressed at high levels in the brain, specifically in neurons, but not in glia, and correlates with neuronal maturation [49,50]. The MeCP2 recognizes and binds to DNA that contains single 5-mC sites. Additionally, the binding of MeCP2 to DNA further recruits transcriptional corepressor complexes, such as Sin3a and histone deacetylases (HDACs) 1 and 2 [47]. During neuronal activity, the activity of MeCP2 often results in the release of promoters due to the removal of the methylation mark on the DNA [51]. Posttranslational modifications of MeCP2, such as its phosphorylation, affect its ability to bind to DNA and alter gene expression [52,53]. Inhibition of MeCP2 phosphorylation is associated with improved synapse formation, synaptic plasticity and learning and memory behavior [54,55]. This is partly because activity-dependent phosphorylation may release MeCP2 from promoters, thereby making the gene sequence available for demethylation process. Furthermore, approximately 95% of cases of neurodevelopmental disorders, such as Rett Syndrome, are due to MeCP2 mutations [56,57]. In an animal model, a 50% increase in MeCP2 in the brain induces impairments in motor coordination, anxiety, long-term potentiation (LTP), learning and memory [58]. Therefore, DNA methylation followed by the binding of MeCP2 appears to have a critical role in synaptic plasticity and cognitive function.

### 2.2. Post-Translational Modification of DNA-Associated Histone Proteins

Nuclear DNA does not appear in free straight strands. It is highly condensed and wrapped around nuclear proteins in the form of chromatin to accommodate chromosomes in the nuclei [59]. The chromatin consists of the nucleosomes. Each nucleosome core consists of two copies of each H2A, H2B, H3 and H4 proteins as well as 147 base pairs of superhelical DNA wrapped around this octamer [60,61]. These four histone proteins form a central structured globular domain that closely contact with the DNA and a less well-structured amino-terminal tail domain [62,63]. Specific amino acid residues of the N-terminal tail domain of these histone proteins undergo chemical modification through PTMs, including acetylation, methylation, phosphorylation, and sumoylation [64]. This distinctive chemical posttranslational modification influences the overall chromatin structure and recruitment of DNA binding proteins and allows for the loosening or tightening of the chromatin around particular genetic loci, thereby resulting in the activation or repression of specific gene expression [64]. As these PTM can alter gene expression without changing the DNA sequence, they have frequently been termed “epigenetic” modifications, although many changes are not maintained across the cell cycle [65].

#### 2.2.1. Histone Acetylation and Deacetylation

Histone acetyltransferases (HATs) transfer the acetyl group from acetyl-coA to the ε-amino group of lysine residues of histone proteins (Figure 2). Three main families of HATs, general control nonderepressible-5 (Gcn5), MYST, and p300/cAMP response element-binding protein (CREB) binding protein (CBP), share a conserved acetyl-coA binding site and often work in large multi-protein complexes [66,67]. The acetylation of histone proteins affects histone-histone protein interactions between neighboring nucleosomes and also interactions between histones and transcription factors (for references see [67]). These alterations can influence the individual nucleosome structure as well as the higher-order chromatin structure, thereby causing a more open and permissive chromatin environment for transcription. While HATs are often not specific to individual lysine amino acid residues, they fulfill specific functions. The broad catalytic role of HATs means that its localization to the proper genomic loci is essential for its particular function. The non-catalytic domains of HATs direct the protein to the appropriate location [68]. Acetylated histones provide a more relaxed chromatin conformation due to hydrophobicity and reduced electrostatic interactions. Additionally, acetylated histones further recruit bromodomain-containing proteins, which are primarily transcription factors and cofactors, and facilitate gene expression [69]. Among the HATs, CBP/p300 is critically involved in development. Genetic mutations and deletions of either enzyme cause Rubinstein-Taybi syndrome, which is characterized by intellectual disabilities, among other impairments [70]. The elevated expression of p300 mRNA was found in neural tissue and the genetic deletion of CBP or p300 resulted in an open neural tube through the impairment of paired box 3 (PAX3) and the activation of enhancer binding protein 2 (AP2) expression, thereby delaying embryo development [71,72]. Both Gcn5/Myc HATs co-regulate transcriptional programs in neural stem cells (NSCs) that are required for stem cell proliferation and brain development [73]. A number of memory tasks have been shown to enhance specific histone H3 and H4 acetylation immediately after memory performance through the enhanced expression and activity of several HATs (Taf1/Kat4, Gcn5/Kat2a, PCAF, Tip60, p300 and CBP) in rodents (for references see [74,75,76,77]). Recently, specific activators of HATs (SPV106, CSP-TTK21, *etc*.) [78,79] have been developed and used in certain neurological disorders for their beneficial function in augmenting neurodegeneration and cognitive function (for references see [80,81,82]). Thus, well-characterized HAT activators may be useful in treating cognitive dysfunction found in FASD children.

Histone lysine deacetylases (HDACs) are a family of chromatin remodeling enzymes that remove acetyl groups from proteins and restrict the access of transcription factors to the DNA, thereby inhibiting gene expression (Figure 2). There are 11 well-described HDACs that are grouped into the following four classes, based upon their sequence, subcellular distribution and function: Class I (HDAC1, 2, 3, and 8), Class IIa (HDAC4, 5, 7, 9), Class IIb (HDAC6, 10), and Class IV HDACs (HDAC11) [83]. These HDACs are zinc-dependent enzymes. Class I HDACs (1 to 3 and 8) are largely localized to the nucleus, but they can also be found in the cytoplasm. Class II HDACs (4 to 7, 9, and 10) and IV HDAC (11) are either found as nucleocytoplasmic shuttling enzymes or primarily cytoplasmic. Class IIa HDACs (4, 5, 7, and 9) have been shown to exhibit low intrinsic deacetylase activity [84]. A separate family of HDACs, which are referred to as sirtuins, are classified as Class III HDACs. These are nicotinamide adenine dinucleotide (NAD)-dependent enzymes, which have mechanisms of action that are distinct from those of other HDACs [85,86]. Not much is presently known about HDAC11, as more studies have yet to emerge about its role in the adult brain. Class I HDACs were expressed at more elevated levels throughout the adult brain in comparison with many other HDACs [87,88,89]. The precise localization and cell-specific expression of individual HDACs in the brain suggests that each HDAC may have a broader, yet specific, significant role in neuronal development and function. Indeed, studies of a conditional knockout of both HDAC1 and HDAC2 suggest that neuronal progenitors fail to differentiate into mature neurons and lead to cell death, thereby resulting in animal death by Postnatal Day 7 (P7) [90]. Additionally, HDAC2 was also critically required to suppress progenitor transcripts during neuronal differentiation of adult-generated neurons [91]. Furthermore, HDAC 1/2 support the stability of synaptic networks during early brain development by suppressing the maturation of excitatory synapses [92]. The genetic deletion of HDAC3 suppresses the translocation of thyroid hormone receptors (SMRT) that regulate the neurogenic pathway to the nucleus [93]. The class II HDAC5 has also been suggested to have an important role in neuronal differentiation [94]. HDAC10 has also been shown to interact with and maintain a deacetylated state of Pax3, a key transcription factor in stem cell maintenance and neurogenesis [95]. Together, these findings suggest that HDAC family members control gene expression and cell fate during brain development into adulthood. Some HDAC families function independently of their catalytic activity function in neurite formation, dendrite and axonal growth (for references see [96]). Methods that involve pharmacological (HDAC inhibitors) and genetic manipulations of HDACs have shown a role of an individual family of HDACs in learning and memory process (for references see [92]). The conditional, brain-specific genetic deletion of HDAC2 in the mouse brain display enhanced acquisition of associative fear learning, whereas the overexpression of HDAC2, but not HDAC1, in the brain led to deficits in fear learning and aversively motivated spatial memory performance in the Morris water maze task [97]. Administration of an HDAC inhibitor, in addition to improving learning and memory, also enhanced the induction and maintenance of LTP at hippocampal and amygdala synapses, two brain regions that are essential for associative learning [76,98,99,100]. It should be noted that many learning and memory studies have utilized trichostatin (TSA) or suberoylanilide hydroxamic acid (SAHA) as an HDAC inhibitor. In doing so, researchers have observed that they appear to exhibit specificity, as these drugs act on class I, but not class II, HDACs [101]. The preclinical use of HDAC inhibitors as therapeutic agents was effective in many animal models of neurological disorders and the study of the benefits of HDAC inhibition for treating several cognitive disorders is underway [102]. Therefore, well-characterized HDAC inhibitors may be valuable for the improvement of the cognitive function of FASD children.

#### 2.2.2. Histone Methylation and Demethylation

The methylation of histones usually occurs at the arginine or lysine N-terminal region of histones, which causes multiple effects, including the activation or inhibition of gene transcription. Each arginine residue in histone proteins can be subjected to monomethylation and symmetric/asymmetric dimethylation. Similarly, each lysine residue can be monomethylated, dimethylated and trimethylated and also adjacent residues can form different methylation combinations [103,104,105,106]. Arginine residue methylation is catalyzed by the family of arginine methyltransferases (PRMT). PRMT1 catalyzes the monomethylation and asymmetric dimethylation of arginine residues, thereby resulting in gene activation. PRMT2 catalyzes monomethylation and symmetric dimethylation of arginine residues, which results in gene suppression. Furthermore, the cofactor-associated arginine methyltransferase (CARM1) catalyzes the formation of monomethylation and asymmetric dimethylation of arginine residues, which promotes gene expression.

Histone methyltransferases (HMTs) that methylate not only lysine amino acids of histone proteins but also other proteins, are called lysine methyltransferases (KMTs) [107] for their broad enzymatic activity and substrate specificity (Figure 3A). All KMTs, except for KMT4, contain a conserved Su (var)3-9, Enhancer of Zeste, Trithorax (SET) domain that exhibits catalytic activity (for references see [108]). The presence of a characteristic homologous sequence [107,109] has helped to classify KMTs into distinct subcategories. Lysine methylation of histone H3K4 is catalyzed by a family (KMT2) of mixed lineage leukemia (MLL) proteins, such as SET1A, SET1B, MLL1, MLL2, MLL3, MLL4 and ASH1 [110,111,112]. H3K4 methylation closely correlates with the activation of gene expression. H3K9 methylation is catalyzed by a group of methyltransferases (KMT1: SUV39H1, SUV39H2, Eu-HMTase/GLP, G9A and ESET/SETDB1; KMT8:RIZ1), whose catalyzing substrates and resulting products are different. The trimethylation of H3K9 is catalyzed by SUV39 and results in the formation of heterochromatin and transcriptional inhibition [113]. The dimethylation of H3K9 is catalyzed by G9a and results in a euchromatin structure and the suppression of gene expression [6,114]. G9a-like protein (GLP) can form a hetero polypeptide complex with G9a, and together they catalyze the formation of H3K9 dimethylation [115]. ESET/SETDB1 catalyze the trimethylation of H3K9 enzymes [116], which results in gene inhibition. H3K27 methylation is mediated by the EZH2 enzyme (KMT6) and results in the inhibition of gene transcription and has been shown to have a significant role in fetal development and cell differentiation [117,118]. H3K36 methylation activity is performed by (KMT3) SET2, SYMD2 and NSD1 enzymes and is associated with active transcription [119,120,121]. H3K4 methylation by SYMD1-3 enzymes is related to gene activation [122]. H3K79 di- and tri-methylation are performed by DOT1L enzymes (KMT4) and are universally associated with active transcription [123]. H4K20 mono- or tri-methylation is catalyzed by SET8 and SUV420H1/2 enzymes (KMT5), respectively, and have been linked with transcriptional repression [124].

Experimental data concerning enzymatic histone demethylation were not known until the discovery of the lysine-specific demethylase 1 (LSD1) (KDM1 family). LSD1 (KDM1A) is characterized as a component of the v-raf murine sarcoma viral oncogene homolog B1 (BRAF)-histone deacetylase (HDAC) (or BHC) complex that contains the restrictive silencer factor (REST) co-repressor, CoREST. CoREST has been shown to function as a repressor of neuronal genes in non-neuronal cells [125,126,127]. LSD1, through its amine oxidase domain, has been shown to act as a transcriptional repressor and to demethylate K3K4, thereby providing the first evidence that histone methylation is a dynamic and reversible event [128]. LSD2 (KDM1B) was recently described as an additional H3K4 demethylase [129]. Similar to KDM1A, KDM1B was shown to demethylate the mono- and di-methylated histone H3K9 [130]. Soon after discovery of the KDM1 family, a protein containing a Fe^2^^+^ dioxygenase Jumonji-C (JmjC) domain (KDM2 family proteins) was characterized as a specific H3K36me1/2 demethylase (KDM2A and B), which does not demethylate H3K36me3 [131]. Now, accumulating evidence suggests the presence of specific demethylases for all of the major methylated histones that have so far been studied (Figure 3B), with the exception of methylated histone H3K79, despite its characterization as a reversible process [123]. The KDM3 family of proteins was characterized as the second family of JmjC histone demethylases (KDM3A/JHDM2A and KDM3B/JHDM2B) and was shown to demethylate mono- and di-methylated H3K9, thereby resulting in active gene expression [132]. The KDM4 family proteins consist four histone demethylases (KDM4A–4D or JMJD2A–2D, respectively). All KDM4 demethylases remove di- and tri-methylation from histone H3K9 and H3K36 proteins [133,134,135]. In addition to the JmjC domain, they all contain a conserved JmjN domain. All, except KDM4D, have tandem plant homeodomain (PHD) fingers and Tudor domains that help to recognize specific histone methylation marks [136,137]. These families of proteins were activated in response to hormonal activity. For example, KDM4A and KDM4D act as androgen receptor (AR) coactivators: estrogen receptor (ER) activation results in KDM4B-mediated demethylation of H3K9me3 proteins [138,139]. KDM5 (KDM5A–5D/JARID1A–1D) specifically catalyzed the demethylation of histone H3K4me2/3 proteins [140,141,142,143,144]. These proteins are characterized as multi domain-containing proteins and include a combination of JmjC/JmjN catalytic domains, with an AT-rich interaction domain (ARID) DNA-binding domain, a C5HC2 zinc finger and two to three PHD fingers. The PHD finger domains of KDM5A and KDM5C were shown to facilitate binding to methylated H3K4 or H3K9, respectively [141,145]. KDM6 family proteins contain KDM6A/UTX and KDM6B/JMJD3 demethylases. These enzymes catalyze the removal of di- and tri-methyl groups form histone H3K27me2/3 proteins [146,147,148,149,150] and result in transcriptional activation. KDM6A functions in development via its association with the KMT2/MLL proteins to regulate H3K4 methylation followed by HOX gene expression [149]. The KDM7 (PHF2) family enzymes contain three members (KDM7A/JHDM1D, KDM7B/PHF8, and KDM7C/PHF2). This enzyme family removes mono and di methyl group from histone H3K91/2 and H3K27me1/2 proteins [139,151,152,153,154,155,156,157]. Additionally, they regulate the expression of genes involved in X-linked mental retardation and expression of ribosomal RNA [154,155,157]. Additionally, PHF8 demethylates histone H4K20me1 proteins [155,157,158]. KDM7 family proteins have a PHD finger domain that helps to recognize and bind to histone the H3K4me3 mark. This ensures substrate specificity, genomic occupancy, and the regulation of target gene expression by KDM7 demethylases [152,156,159,160,161]. Few studies have explored the participation of histone demethylase in developmental alcohol studies, except in one study in which acute prenatal alcohol exposure at GD7 was shown to disrupt the transcripts that encoded KDM1A and KDM4A proteins [162] (see Section 3 for details).

### 2.3. MicroRNAs

MicroRNAs (miRNAs) are a family of small, non-coding RNA molecules that facilitate the degradation of untranslated regions of target mRNAs. Thus, miRNAs play a critical epigenetic function by inhibiting the translation and transcription of a network of genes encoding for proteins. MicroRNAs are generated from longer precursor miRNA (pre-miRNAs) in sequential biogenesis pathways (Figure 4). MicroRNA genes primarily bind to the imperfect complementary sequence that forms a secondary stem-loop structure within the pre-miRNAs. This structure, along with its guide strand [163,164], act as a substrate for two double-stranded RNases, Drosha (nucleus) and Dicer (cytoplasmic) [165]. This specific targeting occurs through a partial complementary association between the mRNA’s 3′UTR and 5′ end of the microRNA (a 6–8 nucleotides sequence long) regions. This partial association facilitates the targeting of a single miRNA to many mRNAs at the same time and *vice versa*. This provides a mechanism wherein a single mRNA can be influenced by diverse miRNAs [166,167]. To date, a few miRNAs have been shown to be expressed in mammalian brains and recent estimates suggest the presence of 1100 or more human miRNAs, which regulate the expression of nearly two-thirds of all genes [168]. Thus, they can control the expression of networks of genes and whole pathways and therefore are considered as master regulators of gene expression [169]. Therefore, miRNAs are critically essential for the proper development and functioning of the adult brain. Indeed, many recent studies suggest the involvement of several miRNAs in an array of cellular homeostatic processes, including developmental timing, cellular differentiation, neural patterning, apoptosis and synaptic plasticity [170,171,172,173].

Studies involving individual genetic deletion of miRNA have suggested that miR-124, miR-125b, miR-132, miR-134, miR-137, and miR-138 in neurons regulate synaptic development and dendritic branching (for references see [174]). The disruption of miRNA biogenesis pathways, such as Dicer, which affects the expression of all miRNAs, can induce impaired cell differentiation, reduction in neuronal size, loss of dendritic branching, disturbed axonal guidance [175], impairments in inhibitory synaptic transmission and cognitive dysfunction [176]. It should be noted that the activity-dependent regulation of gene expression is essential for synaptic plasticity and memory formation and similar neuronal activity was shown to regulate miR-132 expression in a CREB signaling pathway-dependent manner [177,178]. In a recent study, reduced miR-125a levels were found in synaptoneurosomes after activation of mGluR1/ 5 receptors, thereby indicating that neuronal activity may influence the miRNA turnover [179]. Additionally, miRNAs in neurons decay much faster than in non-neuronal cells, and their turnover rate is activity-dependent [180]. Therefore, similar to many learning and memory genes, miRNAs can be regulated in a neuronal activity-dependent manner. It has been suggested that neuronal activity is one mechanism for the high turnover rate of miRNAs in neurons. Consistent with their neuronal function, increasing evidence suggests that the activity of miRNAs is directly implicated in the pathogenesis of complex neuropsychiatric disorders (for references see [181]) that are linked to abnormalities in synaptic plasticity as well as in neurodegenerative diseases [182,183]. Limited studies have explored miRNA dysregulation in FASD animal models, as described later in this section. Therefore, an understanding of the signaling mechanisms that regulate the patterns and activity of miRNA expression have the potential to serve as potential targets for FASD treatment.

## 3. Alcohol’s Influence on Epigenetic Mechanisms in the Developing Brain

The presence of fetal alcohol syndrome was recognized after the initial study [184] suggesting series of deformities in the offspring of alcoholic mothers. Among the several possible consequences of developmental alcohol exposure, defects in brain maturation and the considerable lifelong behavioral impairments have been the most catastrophic. The most serious consequences of early developmental alcohol exposure are fetal alcohol syndrome (FAS) [184], which is characterized by pre- and post-natal growth defects, craniofacial anomalies and persistent impairments of CNS function. FAS occur in 1–2 of every 1000 newborns. It has been acknowledged as an important cause of non-genetic intellectual disabilities and behavioral impairments in the world [185,186,187,188,189]. Nevertheless, now, it has become clear that FAS is not the only enduring defect that results from developmental alcohol exposure. Unquestionably, the term fetal alcohol spectrum disorders (FASD) [190] (estimated the prevalence of as high as 2%–5%) has been used to represent the persistent nature of developmental alcohol effects, with FAS on the stronger side of the spectrum. Though the behavioral profile of FASD is somewhat limited [191], the available literature suggests that FASD is associated with many neuropsychological impairments, such as verbal learning/recall abilities, learning and memory [187,192,193,194,195,196]. These neuropsychological deficits create many daily challenges for individuals with FASD and are also responsible for causing a majority of intellectual disability in the Western world [194,196,197,198]. The neurobehavioral outcome of FASD is also dependent on the amount [199,200], pattern (continuous *vs.* binge drinking) and developmental timing [201] of alcohol exposure. Additionally, the pattern of alcohol exposure can often undermine alcohol dose effects. For example, binge-like exposures result in more severe behavioral outcomes than does chronic exposure [202,203]. Furthermore, several other factors, such as maternal metabolism, genetic susceptibility, and variation in the vulnerability of different brain regions, may contribute to behavioral outcomes. Nonetheless, factors that alter the peak blood alcohol levels (BALs) experienced by the embryo or the fetus (e.g., maternal metabolism) are expected to affect the occurrence and severity of FASD. Unfortunately, this information is often hard to obtain, mainly in retrospectively recruited samples, and individual studies provide varying degrees of detail concerning the amount and patterns of exposure. These issues highlight the modifying variables that contribute to the vast range of phenotypes presented in FASD children.

Persistent data from all areas of biological science, such as epidemiology, genetics, epigenetics, neuroscience and neuroimaging studies, strongly suggest that many adult neuropsychiatric illnesses have a neurodevelopmental origin [204]. Recent technological advances and experimental evidence have provided more precise mechanistic insights on the developmental origin of adult diseases. Disturbance of brain development at sensitive and critical periods by adverse environmental exposure, along with genetic programming, results in the early or late manifestation of heterogeneous clinical conditions [205,206,207,208,209]. The range of developmental deficits varies from learning or cognitive functions to global impairments of social skills or intellect. To date, the etiological basis of the majority of developmentally related adult diseases, such as autism spectral disorder (ASD), attention deficit hyperactivity disorder (ADHD) and FASD conditions, are still unknown. Data from several studies support their polygenic and multifactorial, including epigenetic etiology [4,6,210,211,212,213].

Using animal models of FASD, several studies have reported epigenetics changes, including DNA methylation [214,215,216,217,218,219], DNA-associated histone modification [4,5,6,213,220,221,222,223,224] and miRNAs [225,226,227,228,229,230,231]. Acute alcohol exposure at Gestational Days (GD) 9 through 11 was found to induce hypomethylation of DNA in the fetus [215], and this study was the first one to indicate that DNA methylation may contribute to abnormalities found in FAS. In the prenatal alcohol model, alcohol was shown to significantly decrease astroglial-specific marker, glial fibrillary acidic protein (GFAP) immunoreactivity and its mRNA levels in both astrocytes in primary culture and brains of pups from alcohol-fed mothers. The reduced GFAP transcript found in fetal brains was due to the hypermethylated state of the GFAP DNA [232]. These findings suggest that alcohol-induced DNA hypermethylation, followed by impaired GFAP expression and astroglial development, may underlie the deficits in brain functions that are observed after prenatal alcohol exposure. In some instances, male alcohol exposure before breeding resulted in reduced cytosine methyltransferase mRNA levels in paternal sperm, thereby suggesting that distorted genomic imprinting caused by DNA hypomethylation leads to the expression of normally silent paternal alleles [233]. This phenomenon was also demonstrated between alcohol use in men and hypomethylation of paternally imprinted loci in sperm DNA in genomic regions that are essential for embryonic development, therefore providing a mechanism for paternal effects in the etiology of FASD [218].

The imprinted genes are expressed mainly from either the paternal or maternal allele and are important regulators of fetal growth and development [234]. Additionally, imprinted genes have been found to cause some birth defect syndromes in humans [235,236] and in animals [237,238,239]. In a whole-embryo culture, alcohol enhanced DNA methylation, particularly in genes on chromosome 7, 10 and X, and a neural tube defect phenotype. Enhanced DNA methylation was found in imprinted genes, genes that are essential for cell cycle, growth, apoptosis, cancer, and in a significant number of genes important for olfaction, and with associated changes in gene expression [217]. In monolayer cultures of NSCs, alcohol was shown to increase DNA methylation of multiple cell cycle genes and increased the expression and activity of DNA methyltransferases [240]. In another study, prenatal alcohol-exposed embryos exhibited decreased DNA methylation at insulin-like growth factor 2 (IGF2) transcripts, which was reversed by methyl supplementation [214]. In NSC culture, alcohol exposure retarded the migration, neuronal formation, and growth processes of NSCs. Additionally, alcohol exposure prevented DNA methylation of specific genes related to neural development, eye development and developmental disorders, thereby suggesting that developmental alcohol alters DNA methylation of genes that are critical for regulating neural development [219]. Furthermore, in studies using a whole-embryo culture, alcohol delayed the DNA methylation program, which paralleled the developmental impediment of the neural tube in a spatial and temporal manner [241]. A delay in hippocampus development was also associated with altered 5-mC and 5-hmC in DG and CA region of the hippocampus in prenatally alcohol-exposed embryos or P7 mice [242]. Although these studies have demonstrated the presence of altered methylation program that correlated well with a developmental delay in the formation of the neuronal tube and the hippocampus, whether the abnormal DNA methylation is responsible for the developmental delay and behavioral deficits remains to be established in further studies. In another animal model of FASD (third-trimester equivalent pregnancy), studies have suggested a direct link of this phenomenon. Alcohol administration in Postnatal Days 2–10 (P2–10) mice significantly caused hypermethylation of DNA in the hippocampus (HP) and prefrontal cortex (PFC) at P21. Administration of choline (P2–20), a nutrient supplement that has been shown in a rat model to rescue some of the alcohol’s behavioral effects [243,244], prevented DNA methylation in alcohol-treated P21 mice HP and PFC tissues [245]. In a gestational alcohol model, alcohol increased the methylation of the proopiomelanocortin (Pomc) DNA in a gene- and cell-specific manner in the adult rat hypothalamus despite the exposure to alcohol having ended during the prenatal period [221]. Interestingly, the hypermethylated Pomc gene was found in the sperm of prenatal alcohol-exposed F1 offspring that was transmitted through the F3 generation through the male germline [221]. Global DNA methylation was found to be reduced by alcohol in an *in vitro* murine embryonic fibroblast (MEF) model. Reduction in DNA methylation was associated with alcohol-induced proteasomal-mediated degradation of DNA methyltransferases (DNMT-1, DNMT-3a, and DNMT-3b) [246]. In a neonatal animal model of FASD (third-trimester equivalent pregnancy), the administration of alcohol to P7 mice resulted in reduced global DNA methylation in neonatal HP and neocortical (NC) tissues [23] and impaired learning and memory in adult animals [4,213,247]. This reduction in DNA methylation was linked with alcohol-induced caspase-3 mediated degradation of DNMT-1, DNMT-3a proteins [23] and impaired learning and memory [4,213,247]. Combined pre-(GD1-21) and postnatal (P2–10) alcohol exposure resulted in a significant increase in DNA methyltransferase activity (DNMT), as well as Dnmt1 and Dnmt3a gene expression in the HP of P21 rats [248]. These studies warrant future investigation to understand how these events affect global or gene-specific DNA methylation and learning and memory. Prenatal alcohol exposure decreased DNA methylation in the solute carrier family 17 member 6 (SLC17A6) promoter region, which encodes vesicular glutamate transporter 2 (VGLUT2). In contrast to Slc17a6 mRNA levels, HP VGLUT2 protein levels were significantly reduced in adult alcohol-exposed offspring and suggest that DNA methylation-regulated glutamate neurotransmission in the hippocampus could contribute to the cognitive and behavioral phenotypes that are observed in FASD [249]. Another important observation was made using gestational alcohol exposure in epigenetically sensitive alleles (metastable epialleles) in the Agouti viable yellow (A^vy^) mice model [250]. The activity of A^vy^ is changeable among genetically identical mice, thereby resulting in mice with a range of coat colors [from yellow to mottled to agouti (termed pseudoagouti)] [251]. In this model, exposure to alcohol increases the probability of transcriptional silencing of the A^yy^ locus through DNA methylation, thereby resulting in more mice with an agouti-colored coat [35]. A genome-wide DNA methylome analysis revealed widespread alcohol-induced alterations with significant hypermethylation of many differentiation-related genes in human embryonic stem cells (hESCs) [252]. In a Japanese rice fish embryogenesis model, alcohol was shown to have the capacity to specifically alter the developmental pattern of DNA methylation machinery [253]. A genome-wide analysis of DNA methylation in adult mice exposed to prenatal alcohol using voluntary alcohol exposure paradigm indicated that a large number of gene promoter regions are differentially affected. Furthermore, network analysis of these promoter regions has suggested that behavior-, neurological disease-, and psychological disorder-associated networks are the most significantly affected ones [254]. Enrichment analysis using the genes that were up-methylated in FASD children from 5 to 18 years of age showed a significant enrichment in genes associated with neurodevelopment and diseases [255]. It is clear from the above-discussed studies that the alteration of global DNA methylation or methylation (Table 1) of specific genes/promoter regions is dependent upon several factors, including the specific gene or tissue, amount and pattern of alcohol exposure, as well as the developmental timing of exposure. DNA methylation or demethylation (5-hCm) can influence gene expression through enhancing or preventing the recruitment and binding of transcriptional regulatory proteins with decreased [256] or increased [257,258] transcription activity. Findings also suggest that increased DNA methylation in the gene body can lead to an increased transcription, thus implicating a bidirectional regulation of transcription, depending upon the site of the DNA methylation mark [259]. The mechanisms of the interplay between DNA methylation and transcription activity are therefore crucial. Furthermore, the signaling events that are responsible for DNA methylation-related chromatin remodeling are largely unknown and warrant future research in this line to develop potential therapeutic approaches to treat FASD.

Limited laboratories using many animal models of FASD have demonstrated the participation of DNA-associated histone proteins modifications in responses to alcohol exposure during the different sensitive period of brain development. The earliest report suggests that binge-like alcohol exposure reduces the level of histone H2A family proteins in endothelial caveolae cells [266]. Another study demonstrated the reduced expression of the CREB binding protein (CBP), a histone acetyltransferase, in the rat cerebellum exposed to alcohol during the third trimester-equivalent of human pregnancy. These studies also demonstrated the global reduction of both acetylated histone H3 and H4 proteins in the cerebellum [222]. In a prenatal alcohol model, alcohol was shown to affect the hypothalamus function, such as stress axis responsiveness, which may be due to reduced expression of Pomc gene and production of β-endorphin protein in the arcuate (ARC) nucleus of the hypothalamus. In the same arcuate nucleus, reduced staining of H3K4me2,3, acetylated H3K9 and increased H3K9me2 was also observed in alcohol-exposed male offspring. Consistent with acetylated or methylated histone H3 and H4 staining data, the transcriptional activity of enzymes responsible for methylation or acetylation of histone H3 and H4 proteins (G9a, Set7/9, Setdb1 and HDAC2) were also altered in alcohol-exposed offspring [221]. In another study, during the third-trimester equivalent human pregnancy animal model, alcohol was shown to enhance the expression of G9a mRNA and protein in NC and HP tissues. This resulted in enhanced dimethylation of H3K9 and H3K27 followed by caspase-3 mediated degradation of these proteins. Further studies using G9a (Bix 01294), as well as caspase-3 (Q-VD-Oph) inhibitors, have suggested that the activation of G9a results in robust neurodegeneration in neonatal mice and deficits in synaptic plasticity and learning and memory in adult mice [6,213] as observed in FASD children [193,195,198]. These neurobehavioral impairments were rescued by preventing the alcohol activated G9a [213]. Similarly, alcohol also enhanced the transcriptional activity of cannabinoid receptor 1 (CB1R) through promoter (exon1) specific enrichment of H4K8ace protein in the mouse NC and HP. Transcriptional activation of CB1R results in impaired CB1R-mediated signal transduction (pERK1/2/pCREB/activity-regulated cytoskeletal protein (Arc)), resulting in neurodegeneration in neonatal mice and learning and memory deficits in adult mice [4,247]. These neurobehavioral deficits were prevented by CB1R antagonist or CB1R knockout mice [4,247]. Furthermore, a promoter specific analysis of histone H3 proteins using low dose alcohol (elicit mild neurodegeneration) showed an enhanced specific acetylation of H3K14 and no change in H3K9me2 or H4K8ace levels, as observed in the G9a exon1 region [5]. Transcriptional activation of G9a results in increased expression of the G9a gene and protein, thereby resulting in enhanced global dimethylation of H3K9 and H3K27 proteins without any significant degradation of H3 proteins due to the low activity of caspase-3, as found in a low-dose alcohol model [5]. In another prenatal alcohol model, alcohol was shown to enhance the expression of genes that have a role in the fetal heart through promoter-specific H3K14 acetylation in mice [267]. In a similar fetal heart study, alcohol increased global HAT activity without affecting global HDAC activity. Additionally, alcohol increased the binding of P300, CBP, PCAF, SRC1, but not GCN5, at the Gata4 promoter region in comparison to the saline-treated group. The increases in acetylation of H3K9, as well as in Gata4, α-myosin heavy chain (α-MHC) and cardiac troponin T (cTnT) mRNA levels, were also evident in these fetal hearts [268]. Studies in a prenatal alcohol model suggest that alcohol enhanced the enrichment of histone H3K4me3 at the Slc17a6 promoter, which encodes VGLUT2 [249]. Acute exposure to alcohol at GD 7 in mice exhibit significantly increased H3K9me2 and H3K9ace levels and reduced H3K27me3 levels at GD 17 and that these histone modifications strongly correlate with the development of craniofacial and CNS defects in this animal model. These observations indicate that an epigenetic signature persists long after the window of alcohol exposure and is mostly linked to alterations in a repressive chromatin structure, such as H3K9me2 [162].

The first evidence of histone acetylation modification by developmental alcohol was found in cardiac progenitor cells where alcohol enhanced H3K9ace and augmented the mRNA expression of heart development-related genes [224]. Exposure to alcohol during the third trimester-equivalent of human pregnancy reduces CBP levels, followed by reduced histones H3 and H4 acetylation in the neonatal cerebellum [222]. Prenatal alcohol exposure enhanced long-lasting H3K9ace marks and CBP mRNA levels in β-endorphin (BEP)-positive cells in the ARC of the hypothalamus of male rat offspring [221]. The transient exposure of immature P7 mice to alcohol, which is comparable with a time point within the third trimester of human pregnancy, enhances specific acetylation of H3K14 in the G9a exon1 gene, which results in enhanced G9a transcription [5]. In a similar mouse model, it was shown that alcohol enhances acetylation of H4K8 in the CB1R exon1 gene, thereby leading to enhanced CB1R expression and impaired synaptic plasticity, learning and memory in adult mice [4].

Prenatal alcohol exposure reduced both mRNA levels of KMT7/9 (Set7/9) and the number of H3K4me2, 3-positively stained BEP cells. Additionally, enhanced KMT1C (G9a) and KMT1F (Setdb1) mRNA, as well as H3K9me2-positively stained BEP cells, was observed in the ARC of the hypothalamus of rat offspring exposed to prenatal alcohol [221]. In another study, binge-like alcohol exposure in P7 mice, which is comparable with the third trimester of human pregnancy, enhanced KMT1C (G9a) expression, as well as in H3K9me2 and H3K27me2, in both neocortical and hippocampal tissues of neonatal mice [5]. This particular dimethylation of both histones marks them for proteolytic degradation by activated caspase-3 in this animal model [6]. The inhibition of KMT1C (G9a) activity before alcohol treatment not only rescued neurodegeneration in neonatal mice but also synaptic plasticity and learning memory in adult mice [213]. Gestational choline treatment, which has been shown to prevent pre/postnatal alcohol-induced learning and memory [244,269], also normalized the alcohol-suppressed hypothalamic levels of H3K4me3, as well as the NA levels of KMT7/9, and increased the H3K9me2 and mRNA levels of KMT1C and KMT1F in the hypothalamus of adult male offspring rats [220]. In another study, prenatal alcohol exposure enhanced SLC17A6 gene expression, which codes for VGLUT2 through the enrichment of H3K4me3 at the SLC17A6 promoter region in adult mice [249]. In another study, acute exposure of GD7 mice to alcohol significantly disrupts transcripts that encode for KMT1C (G9a), KMT6 (EzH2) and KMT7 (SETdB1) proteins [162]. Together, these findings suggest that these epigenetic mechanisms (Table 2) may have a direct role in cognitive dysfunction observed in FASD.

The first evidence to show that early alcohol alters miRNA (Table 3) was shown in NSCs using *ex vivo* fetal cerebral cortical neuroepithelium model [228]. In this study, alcohol, which was shown to promote cell proliferation and aberrant differentiation, did not cause the death of NSCs, but rather suppressed miR-9, -21, -153, and -335 levels [228]. Furthermore, it has been shown that the lack of cell death by alcohol in this study was due to the combined loss of miR-335 and miR-21 and the loss of miR-335 alone was responsible for the increased NSC proliferation in response to alcohol [228]. In another study using a prenatal mouse model, it has been shown that alcohol increased certain miRNAs (miR-10a, -10b, -9, -145, -30a-3p and miRNA-152) and decreased other miRNAs (miR-200a, -496, -296, -30e-5p, -362, -339, -29c and -154) in the fetal brain. This resulted in impaired task acquisition in the Morris water maze in P35 mice [231].

In another study, Zebrafish embryos were exposed to alcohol, which was then followed by the evaluation of miRNA levels and deformation. This study showed increases in miR-153a, -725, -30d, let-7k, -100, -738, and -732 and decreases in miR-128 and -193b and induced cell death along the dorsal axis of the embryo [229]. In another study, using P7 mouse model of FASD, alcohol was shown to enhance miR-26b levels in adult whole brain tissues [230]. MicroRNA-26b, which is implicated in the differentiation of NSCs [227], has also been shown to control the expression of genes that encode for brain-derived neurotrophic factor (BDNF), which is important for neurodevelopment [226,227]. Exposure of GD12.5 mouse fetal murine cerebral cortical-derived neurosphere cultures to alcohol suppressed the expression of several miRNAs, including miR-9, -21, -153, and -335, as well as miR-140-3p [225]. In Zebrafish embryos, alcohol suppresses miR-9 biogenesis. This early embryonic loss of miR-9 function recapitulates the severe range of teratology that is associated with early alcohol exposure [270]. In another study that uses an acute prenatal model (second-trimester equivalent), alcohol was shown to enhance miR-302c in the adult whole brain [271]. MicroRNA-302 is suggested to have a critical function in neural tube closure and embryonic survival [274]. Furthermore, rats treated with acute alcohol at GD 12 exhibit both upward and downward changes in several miRNAs in the amygdala and ventral striatum in association with decreased social motivation behavior at P42 [272]. Social enrichment of GD12 alcohol-exposed rats rescued both miRNA and social motivation behavior in P42 mice [272]. Furthermore, it has been shown that miR-125b can rescue alcohol-induced apoptosis in NCCs by decreasing the expression of Bcl-2 antagonist killer 1 (BAK1) and p53-upregulated modulator of apoptosis (PUMA) genes (pro-apoptotic genes). Additionally, the microinjection of miR-125b mimic into cultured mouse embryos prevented alcohol-induced embryotoxicity [275]. Prenatal exposure of mice on GD1-8 resulted in enhanced miR-135a, -135b, -467b-5p and -487b in the P87 murine hippocampus. Additionally, miR-467b-5p and the 3′UTR of Slc17a6 exhibited specific interactions *in vitro* and suggested that a similar interaction *in vivo* may contribute to reduced HP VGLUT2 protein levels in alcohol-exposed adult males [249]. In this model, alcohol not only enhanced HP levels of miRNAs but also increased miRNAs, miR-135a, miR-135b and miR-467b-5p levels in the serum [249]. These findings suggest that miRNA may have an important role in HP dysfunction found in FASD and circulating miRNAs could be used as biomarkers of early developmental alcohol exposure [249]. A similar change in circulating miRNAs (miR-9, -15b, -19b, and -20a) was also found in pregnant ewes and in newborn lambs (GD 147) that were exposed to alcohol between GD 4 and GD 132 [273]. Alcohol exposure in P7 mice was shown to alter several miRNA transcripts and was found to be associated with genes that are important for neural connectivity [276] and may contribute to cognitive deficits, as found in FASD children.

## 4. Conclusions

In summary, several epigenetic mechanisms are suggested to contribute to the harmful consequences of alcohol abuse during pregnancy in the developing fetus (Figure 5). Epigenetic modifications, such as altered DNA methylation, specific histone protein modification and dysregulation of miRNA, in response to developmental alcohol exposure, can contribute to impaired neurogenesis, neuronal communication and neural circuit assembly. These alcohol-induced neuronal deficits can be long-lasting and could result in abnormalities in synaptic plasticity and cognitive function and can provide bases for many of the neurobehavioral abnormalities found in FASD. The broad role of DNA methylation and chromatin modification in learning and memory has not been fully understood. This is because learning and memory is regulated by a dynamic interplay between the various signaling molecules, enzymes and proteins that are capable of modifying the chromatin structure. Additionally, there is a lack of understanding of the mechanisms that both predicate and result from epigenetic modifications that regulate learning and memory process. Modulation of epigenetic marks using pharmacological and gene knock-in or knockout approaches support a role for the epigenetic process in learning and memory, therefore making it a promising target for the future development of potential therapeutic agents to treat FASD.

## Figures and Tables

**Figure 1 brainsci-06-00012-f001:**
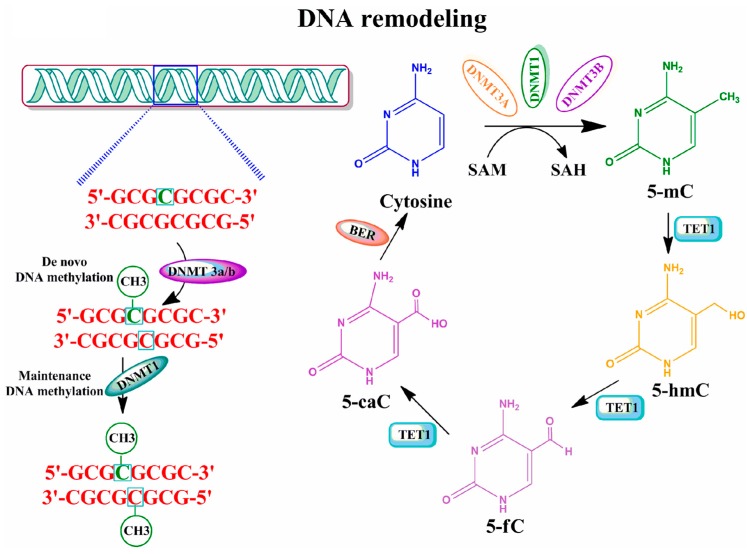
Graphic representation of DNA methylation and its regulation by enzymatic mechanisms. Methylation of DNA begins with the covalent addition of a methyl group from s-adenyl methionine (SAM) [41] to the fifth carbon of the cytosine pyrimidine ring to form 5-methylcytosine (5-mC), a process that is catalyzed by a family of DNA methyltransferases (DNMTs). The majority of DNA methylation usually occurs at genes on cytosines that precede a guanine nucleotide or CpG islands. *De novo* methyltransferases (e.g., DNMT3a/b) transfer methyl groups to naked DNA CpG pairs (e.g., CpG/GpC to mCpG/GpC) [42,43]. DNMT1 is the maintenance methyltransferase that transfers methyl groups to hemimethylated DNA strands (e.g., mCpG/GpC to mCpG/GpCm) and maintains the parental DNA methylation pattern during replication [44]. 5-mC undergoes sequential oxidation to 5-caC by TET1 activities. 5-caC, through base-excision-repair (BER) mechanisms, results in the regeneration of cytosine [39,45,46]. 5-methylcytosine (5-mC); 5-hydroxymethylcytosine (5-hmC); 5-formylcytosine (5-fC); 5-carboxylcytosine (5-caC).

**Figure 2 brainsci-06-00012-f002:**
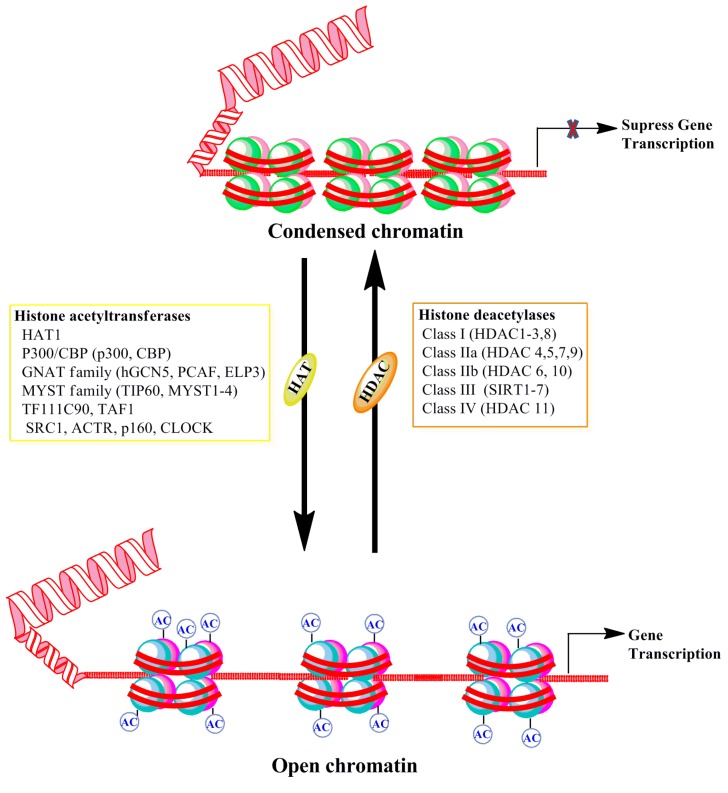
Schematic representation of DNA-associated histone protein acetylation and deacetylation by histone acetyltransferase (HAT) and histone deacetylase (HDAC) enzyme families. The net acetylation at lysine residues of histone proteins of nucleosomes is decided by the interplay between acetylation and deacetylation processes through HAT and HDAC enzyme activities, respectively. The box provides different families and classes of HAT and HDAC enzymes. CBP, cyclic adenomonophosphate response element-binding (CREB) binding protein; GNAT, Gcn5-related *N*-acetyltransferases; hGCN5, human general control of amino acid synthesis protein 5-like 2; PCAF, p300/CBP-associated factor; ELP3, elongation protein 3; TIP60, TAT interacting proteins 60; TFIIIC90, transcription factor IIIC 90kDa; TAF1, TATA Box Binding Protein-Associated Factor; SRC1, steroid receptor coactivator 1; ACTR, activator of thyroid receptor; p160, receptor coactivators proteins 160; CLOCK, Clock Circadian Regulator.

**Figure 3 brainsci-06-00012-f003:**
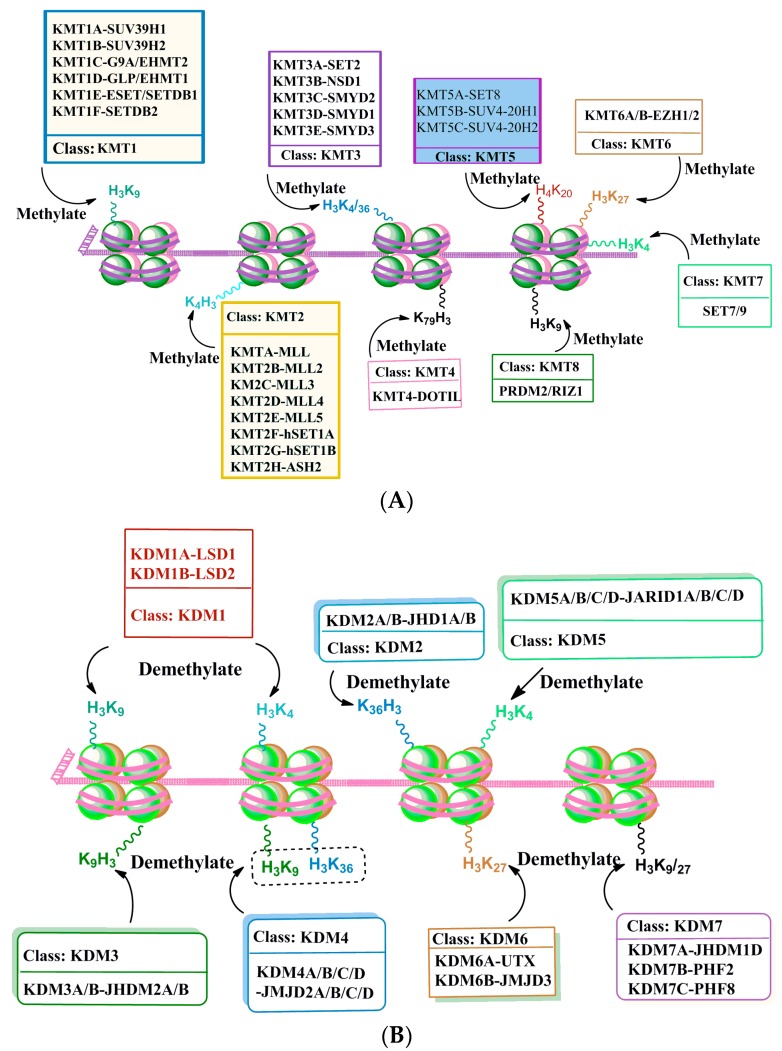
DNA-associated histone protein lysine methylation (**A**) and demethylation (**B**) by histone methyltransferase (KMTs) and histone demethylase (KDMs) enzyme families. Histone H3 and H4 tails with known lysine methyltransferases (KMT1-8) and demethylases (KDM1-7) are shown under each specific lysine residue.

**Figure 4 brainsci-06-00012-f004:**
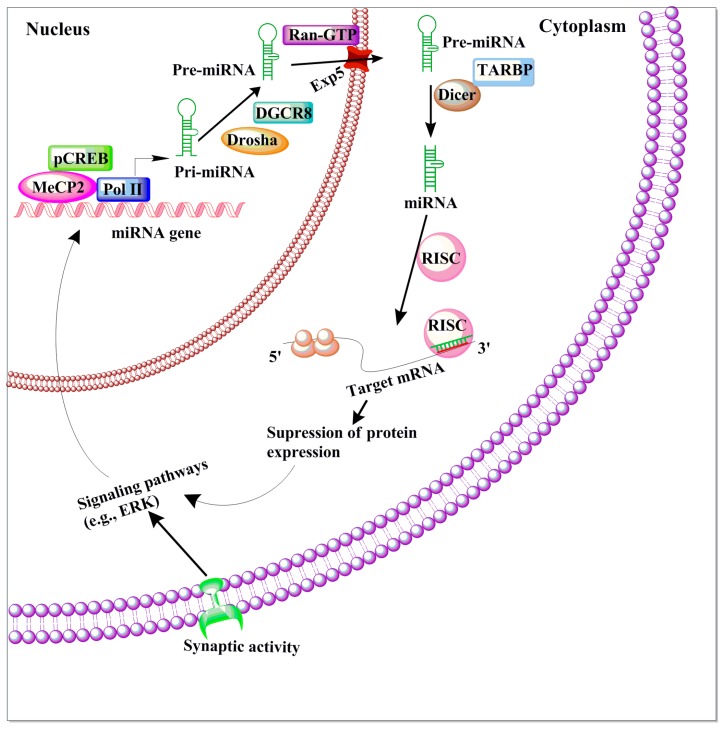
The schematic diagram of miRNA biogenesis and function. In the nucleus, RNA polymerase II (Pol II)-dependent transcription of a miRNA-encoding gene, which may include both intron- and exon-coding regions, results in the formation of a long primary miRNA transcript (pri-miRNA) that is 50-capped and 30-polyadenylated. This pri-miRNA transcript is subject to nuclear processing by the microprocessor complex, which includes DCGR8 and Drosha, into the precursor miRNA (pre-miRNA) transcript. Pre-miRNAs are transported out of the nucleus to the cytoplasm by Exp5 and Ran-GTP. Pre-miRNA can then be further cleaved by Dicer/TARBP to generate a mature miRNA. The mature miRNA is incorporated into the RNA-induced silencing complex (RISC), which will select mRNA transcripts for the down-regulation of protein expression. In the synapses, miRNA can regulate the local down-regulation of protein expression. Certain conditions, such as neuronal activity, could affect the signaling events as well as miRNA formation. Additionally, genes regulated by miRNA can act on synaptic activity-dependent signaling pathways that promote the activation of epigenetic factors (e.g., CREB and MeCP2), which in turn can control miRNA transcription in the nucleus.

**Figure 5 brainsci-06-00012-f005:**
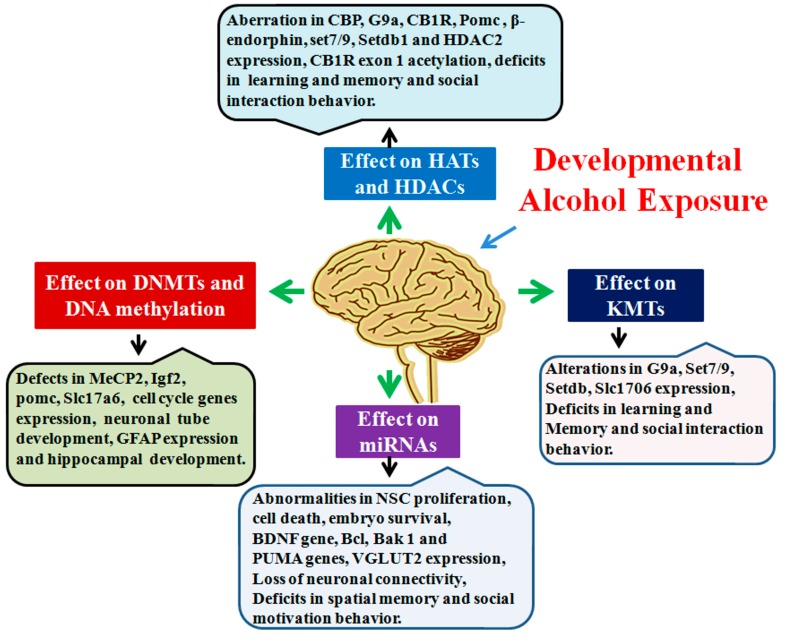
Graphical summary of developmental alcohol-induced epigenetic defects. Developmental alcohol exposure has been shown to affect DNA methylation, HATs/HDACs, KMTs and miRNAs, followed by several changes in genes and protein expression that are important for cognitive and other brain function.

**Table 1 brainsci-06-00012-t001:** Summary of the developmental alcohol-induced DNA methylation modifications.

Alcohol Exposure	Tissue Examined	Effects
1. GD 9–11	GD 12 fetus	Reduced DNA methylation [215]
2. GD	Cultured astrocytes	BDNF gene hypermethylation [260]
3. GD 1–18	PD 10	Hypermethylation of the GFAP gene [232]
4. *Oryzias latipes* embryos	2–6 dpf	No change in *Aldh1A2* gene promoter methylation [261]
5. Embryo (E8.25) cultures	Embryo cultures	Both hyper/hypomethylation of gene promoter [217]
6. Neural stem cells in culture	NSCs culture	Hypermethylation of cell cycle genes [240]
7. Neural stem cells in culture	NSCs culture	Hypomethylation of NSC genes [219]
8. GD 9	E-15 and 17	Decreased *Igf2* gene DNA methylation [214]
9. PD 2–10	PD 21	Increased DNA methylation [245]
10. GD 1–22	PD 21	Enhanced DNMTs activity [248]
11. hESC	Cell lines	No change in gene methylation [262]
12. hESC	Cell lines	Hypermethylation of many regions of chromosomes [252]
13. GD 1–17	Cultured neural progenitor cells	Disrupt DNA methylation machinery and delays the maturation of dentate gyrus [242]
14. Murine embryonic fibroblasts	Fibroblasts	Impaired DNA methylation and DNMT1, DNMT3a and DNMT3b proteins [246]
15. GD 7–21	PD 60–65	Increased DNMT1protein and *Pomc* gene methylation [220]
16. PD 7	PD 7	Reduced DNA methylation, DNMT1and DNMT3a proteins [23]
17. GD 0.5–8.5	PD 28	Decreased mRNA levels of *Dnmt1* and *Dnmt3a* genes [263]
18. GD 0.5–8.5	PD 28	Both up/down regulation of DNA methylation [264]
19. GD 0.5–8.5	PD 87	Decreased Slc17a6 gene promoter [249]
20. *Oryzias latipes* embryos	2–6 dpf	Altered DNMT1 mRNA [265]

dpf, day post fertilization; GD, gestational day; PD, postnatal day; NSCs, neuronal stem cells; GFAP, glial fibrillary acidic protein; *Aldh1A2*, *a*ldehyde dehydrogenase 1 family, member A2; *Igf2,* Insulin-Like Growth Factor 2; DNMT, DNA methyltransferase; hESC, human embryonic stem cells; Pomc, Proopiomelanocortin; *Slc17a6,* Solute Carrier Family 17 (Vesicular Glutamate Transporter), Member 6.

**Table 2 brainsci-06-00012-t002:** Influence of developmental alcohol on histone modifications.

Alcohol Exposure	Tissue Examined	Effects
1. Gestational Day (GD) 7	GD 17	Increased H3K9me2, H3K9ace and decreased H3K27me3 [162] Altered G9a, Setdb1, Kdm1a, Kdm4c, Uhrf1, Ezh2 and Dnmt1 mRNA levels [162]
2. Postnatal Days (PD) 2–12	PD 2–12	Decreased AcH3, AcH4, H3K23ace and increased HAT (CBP) [222]
3. GD 7–21	PD 60–80 from F1–F3 generation	Decreased H3K4me2, H3K4me3, H3K9ace, pH3S10 and mRNA Set7/9 [220]; Increased mRNA G9a, setdb1 H3K9me2 [220] Decreased H3K4me2,H3K4me3, H3K9ace and Decreased mRNA levels of Set7/9; Increased G9a mRNA H3K9me2 and HDAC2 [221]
4. GD 1–14.5	Embryonic days (ED) 7.0–14.5	Increased H3K14ace [267]
5. PD 7	PD 7	Increased H4K8ace [4], H3K14ace [5] Decreased H3K9me2, H3K27me2 [4] Decreased H3K9me2, H3K27me2 [6], H3K9me2 [213] Increased G9a mRNA and Protein [6]
6. ED 8.5–E16.5	ED14.5–PD 7	Increased p300 and SRC1 protein. No Change in HDAC. Increased HAT (CBP and PACF) [268]
7. ED 8.5–16.5	ED 14.5–16.5	Increased H3K14ace [267]
8. Days of post cotium (dpc) 0.5–8.5	PD 87	Increased H3K4me3 and Slcl7a6 gene expression [249]

**Table 3 brainsci-06-00012-t003:** Summary of the developmental alcohol elicited changes in miRNAs.

Alcohol Exposure	Tissue Examined	Effects
1. GD 12.5	Neurosphere cultures	Reduced the expression of miR-9, 21, 335 and -135 [228]
2. GD 6–15	GD 17 embryo culture	Increased miR-10a, -9, -145, -30a-3p and -152. Also decreased miR-200a, -496, -296, -30e-5p, -362, -339, -29c and -154 [231]
3. Zebrafish	Embryos (4–96 hpf)	Enhanced miR-153a, -30d, -736, -183 and reduced -23a [229]
4. GD 12.5	Neurosphere cultures	Reduced the expression of miR-140-3p [225]
5. PD 7	PD 60	Increased expression of miR-26b [230]
6. Zebrafish	Embryos (24–72 hpf)	Suppressed miR-9a and increased the accumulation of pre-miR-9-3 [270]
7. PD 7	PD 60	Enhanced miR-302c [271]
8. GD 1–22	PD 42	Decreased miR-874-5p, 1843a-3p, -221-5p, -29c-3p, -384-5p, -412-3p, 129-1, -138-2, -322-2, -496, -9a-2. Increased miR-155, -34c, -let-7c-1, -let-7c-2-3p, -542-1 [272]
9. GD 0.5–8.5	PD 87	Enhanced miR-135a, -135b, -467b-5b and -487b [249]
10. GD 4–132 (Ewes)	Plasma (GD 147)	Decreased miR-572, -720, -9, -15b, -17-92 and increased miR-34b [273]

hpf, hours post fertilization; GD, gestational day; PD, postnatal day.

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
