# Peer review of "Epigenetic Mechanisms in Developmental Alcohol-Induced Neurobehavioral Deficits"

_brainsci, 2016, doi:10.3390/brainsci6020012_

Round 1
Reviewer 1 Report
Comments to the Author
This review is focused on the role of Epigenetic in Developmental Alcohol-induced Neurobehavioral Deficits. This is a very novel concept with explanatory figures. The text is well structured.
Minor comment:
1. 1: In Figure 5, labeling is unclear. It’s hard to understand.
2. Figure 3 is very complex. Is nice if the author can simply the figure.
3. As Alcohol is the main culprit here it’s nice to see more of alcohol instead of DNA methylation (too large paragraph).
4. The author should incorporate the table of all miRNA which is effect during alcoholism in the brain.
5. In fig.2; Please label the structure before “Suppress gene Transcription” and in bottom “Gene”.
Author Response
Please see attached point-point responses to reviewer comments

Reviewer 2 Report
The review by Basavarajappa and Subbanna “Epigenetic Mechanisms in Developmental Alcohol-induced Neurobehavioral Deficits” submitted for publication in Brain Sciences offers a comprehensive coverage of the current state of our understanding of interaction between developmental exposure to alcohol and genome. Specifically, the consequences of exposure to alcohol during prenatal human development (as studied in animal models) for epigenetic modifications of DNA methylation, DNA-associated histone proteins and micro-RNAs are discussed in depth. My major concern however is the disproportional shift towards the discussion and review of the current literature on DNA methylation and histone proteins modification (10 pages) instead of more in depth discussion of epigenetic changes in FASD (6 pages). I would suggest the authors expand the latter part of their manuscript. In addition, the organization of the factual material is somewhat confusing, especially in the animal models of FASD part of the review. It would be beneficial for the readers to have a clear separation between alcohol effects of epigenetic changes in early embryo vs brain/nervous system vs other systems. Probably, a table could be a good solution for the overwhelming and thus somewhat confusing amount of information.
Minor points:
1. The first sentence in “Alcohol’s influence on epigenetic mechanisms in the developing brain” chapter needs to be re-written, it is grammatically awkward.
2. Same can be said about the third sentence in the same chapter: “The significant effects of early developmental alcohol exposure are fetal alcohol syndrome (FAS)”. The effects on the brain/nervous system can cause or produce FAS or FASD, but they are not FAS themselves.
Author Response
Please see the attached PDF file for a point-point response to reviewer comments

Round 2
Reviewer 2 Report
The authors addressed the concerns of the reviewers and made important changes to the manuscript.